# Sterol O-Acyltransferase Inhibition Ameliorates High-Fat Diet-Induced Renal Fibrosis and Tertiary Lymphoid Tissue Maturation after Ischemic Reperfusion Injury

**DOI:** 10.3390/ijms232415465

**Published:** 2022-12-07

**Authors:** Yuki Ariyasu, Yuki Sato, Yosuke Isobe, Keisuke Taniguchi, Motoko Yanagita, Makoto Arita

**Affiliations:** 1RIKEN Center for Integrative Medical Sciences, Yokohama 230-0045, Japan; 2Department of Nephrology, Graduate School of Medicine, Kyoto University, Kyoto 606-8507, Japan; 3Division of Physiological Chemistry and Metabolism, Graduate School of Pharmaceutical Sciences, Keio University, Minato-ku, Tokyo 105-8512, Japan; 4Cellular and Molecular Epigenetics Laboratory, Graduate School of Medical Life Science, Yokohama City University, Yokohama 230-0045, Japan; 5Institute for the Advanced Study of Human Biology (ASHBi), Kyoto University, Kyoto 606-8501, Japan

**Keywords:** lipidomics, sterol O-acyltransferase, tertiary lymphoid tissue, chronic kidney disease, cholesteryl ester, high-fat diet, metabolic syndrome

## Abstract

Metabolic syndrome is associated with the development of chronic kidney disease (CKD). We previously demonstrated that aged kidneys are prone to developing tertiary lymphoid tissues (TLTs) and sustain inflammation after injury, leading to CKD progression; however, the relationship between renal TLT and metabolic syndrome is unknown. In this study, we demonstrated that a high-fat diet (HFD) promoted renal TLT formation and inflammation via sterol O-acyltransferase (SOAT) 1-dependent mechanism. Mice fed a HFD prior to ischemic reperfusion injury (IRI) exhibited pronounced renal TLT formation and sustained inflammation compared to the controls. Untargeted lipidomics revealed the increased levels of cholesteryl esters (CEs) in aged kidneys with TLT formation after IRI, and, consistently, the *Soat1* gene expression increased. Treatment with avasimibe, a SOAT inhibitor, attenuated TLT maturation and renal inflammation in HFD-fed mice subjected to IRI. Our findings suggest the importance of SOAT1-dependent CE accumulation in the pathophysiology of CKDs associated with TLT.

## 1. Introduction

There is a substantial and increasing burden of obesity on public health worldwide. Obesity is a major risk factor for several pathological conditions, including hypertension, diabetes, cardiovascular diseases, and chronic kidney disease (CKD). Importantly, several clinical studies have demonstrated that obesity is a major risk factor for the development of end-stage renal disease (ESRD), which is independent of diabetes and hypertension [1]. Hsu et al. reported that, in the general population, obese patients (BMI > 30 kg/m^2^) had a 3.57-fold higher risk of developing ESRD compared with normal weight patients (BMI: 18.5–24.9 kg/m^2^) [2]. Experimental studies utilizing a high-fat diet (HFD)-induced obesity model also demonstrated an association between obesity and CKD. In the kidneys of HFD-fed mice, lipid dysregulation, such as accumulation of non-esterified fatty acids and/or cholesterol, was observed, along with increased pro-inflammatory cytokines, oxidative stress, and fibrosis [3,4,5], which are major hallmarks of chronic inflammation. However, despite its clinical significance, the mechanisms by which obesity promotes chronic inflammation remain elusive, and the molecular details of lipid dysregulation in obesity-related kidney disease remain unclear.

CKD prevalence is increasing worldwide, particularly among older adults. A reason for this is that older patients are prone to developing acute kidney injury (AKI), which often leads to CKD development and progression. This “AKI to CKD continuum” has been recognized as one of the most pressing unmet needs in renal medicine [6]. Several causes have been proposed for the AKI to CKD continuum. For example, age-dependent structural changes, such as nephron loss and glomerulosclerosis, and functional impairments, such as decreased proliferation capacity, are recognized as underlying causes of AKI to CKD progression. Additionally, chronic inflammation is recognized as a key player in progression. In a previous study, we demonstrated that aged kidneys, as opposed to young kidneys, are prone to developing tertiary lymphoid tissues (TLTs) after AKI, which leads to maladaptive repair and CKD development [7,8]. TLTs are lymphocyte aggregates that are frequently observed in non-lymphoid organs under chronic inflammatory conditions [9,10]. Similar to the lymph nodes, TLTs have unique structures that allow interaction between T, B, and dendritic cells, and are structurally and functionally supported by specialized fibroblasts within TLTs [11]. TLTs develop and mature through at least three distinct stages [12]. TLTs initially develop as intermingled T and B cell clusters (stage I) in response to injury, and, subsequently, mature into clusters with B cell areas supported by CD21+ follicular dendritic cells (FDCs) (stage II), which are conventional mesenchymal stromal cells for B cell activation and differentiation. TLTs with FDCs have the potential to induce germinal centers, where B cells undergo antigen-driven somatic hypermutations. In the later phase of AKI, certain TLTs in the kidneys harbor germinal centers in B cell areas (stage III). Importantly, the TLT developmental stage is associated with the severity of local tissue injury and inflammation. For example, we recently showed that the presence of stage II TLTs in protocol biopsy is associated with progressive graft dysfunction in kidney transplant recipients [13]. These results suggest that the TLT developmental stage has the potential to be a novel kidney injury marker. Interestingly, TLTs also develop within adipose tissue, termed fat-associated lymphoid clusters, which also promotes T cell proliferation under HFD feeding [14]. However, whether lipid dysregulation contributes to renal TLT formation has been insufficiently investigated.

In this study, we aimed to investigate the relationship between HFD-induced metabolic disorders and renal TLT formation and subsequent CKD development after AKI.

## 2. Results

### 2.1. HFD Feeding Promotes TLT Formation in Aged Kidneys after Ischemic Reperfusion Injury (IRI)

To investigate the effect of HFD feeding on TLT formation in the kidney, we examined aged kidneys subjected to ischemic reperfusion injury (IRI) under control diet or HFD feeding, as depicted in Figure 1A. Body weight significantly increased with HFD feeding (Appendix A). On day 35, after IRI, both groups exhibited mononuclear cell aggregates around the artery, which were primarily composed of T and B lymphocytes with signs of proliferation (Figure 1B,C), indicating that these aggregates function as TLTs. These TLTs also harbored p75NTR-positive fibroblasts, as previously reported [7] (Figure 1C). On the other hand, the contralateral non-injured side of the kidneys did not make any TLTs in both control and HFD-fed mice (Appendix A). Quantitative analysis revealed that the cumulative TLT sizes and total TLT numbers were significantly higher in HFD-fed mice than in control diet-fed mice (Figure 1D). Furthermore, the number and proportion of advanced stage (stage II or III) TLTs were increased, although the difference was not statistically significant (Figure 1D,E). The mRNA expressions of lymphoid homeostatic chemokines, including *Cxcl13* and *Ccl19*, and pro-inflammatory cytokines, such as *Ifng* and *Tnfa*, were also significantly increased in the kidneys of HFD-fed mice (Figure 1F). Additionally, renal fibrosis scores, quantified as the areas of Picrosirius red staining-positive areas, were increased in HFD-fed mice (Figure 1G).

### 2.2. Untargeted Lipidomics Reveals Prominent Increase of Cholesteryl Ester in Aged Injured Kidneys with TLTs

To investigate the metabolic signature associated with TLT formation, we performed untargeted lipidomics of young (8-week-old) and aged (12-month-old) kidneys subjected to IRI by liquid chromatography–tandem mass spectrometry (LC–MS/MS) (Figure 2A). Since both male and female mice over 8 months old exhibit TLTs after IRI, metabolic changes common in both sexes are important. A previous report comparing sex differences in C57B6/J mice fed the same diet showed that there were similar trends in body weight gain, glucose metabolism, blood cholesterol and triglycerides with HFD in both sexes [15]. We utilized male kidneys with different ages because they were previously reported to have marked differences in TLT formation under the same feeding conditions [7]. Aged kidneys, but not young kidneys, exhibited multiple TLTs 45 days after IRI (Figure 2B) [7]. Among the 976 lipids annotated, 511 lipids were found to be significantly different among groups, by one-way analysis of variance. The peak intensities of these lipids were converted to z-scores and entered into hierarchical clustering (Figure 2C). While the majority of the lipids displayed similar behavior associated with injury in both aged and young kidneys, a cluster of lipids was uniquely increased in aged kidneys after IRI. The cluster contained cholesteryl ester (CE) and ether-linked phosphatidylcholine (Figure 2D). Notably, 17 out of 18 CE species levels were increased in aged mice after injury compared to those in young mice (Appendix A), while the levels of 23 out of 42 ether-linked phosphatidylcholine species were increased in the aged injured kidneys (Appendix A). Detailed data of clustered molecules are reported in Appendix A.

### 2.3. Increased Sterol O-Acyltransferase (SOAT) Expression in Aged Kidneys after IRI

Cellular CEs are formed by sterol O-acyltransferase (SOAT) and are hydrolyzed by neutral cholesterol ester hydrolase (NCEH). As CE levels were significantly increased in aged kidneys after IRI (Figure 2E), we next determined the expression levels of SOAT and NCEH enzymes in the kidney. RT-qPCR analysis revealed a significant increase in *Soat1* and *Soat2* mRNA levels in aged kidneys after IRI compared to those in young kidneys, while *Nceh1* remained unchanged (Figure 3A). *Soat1* was expressed over 50-fold higher than *Soat2* in aged injured kidneys, suggesting that SOAT1 is the major enzyme that produces CE in aged kidneys after IRI (Figure 3B).

We performed highly sensitive in situ hybridization to investigate *Soat1* expression in aged injured kidneys with TLTs. *Soat1* is expressed in resident cells in the kidneys, such as tubular and glomerular cells, and in renal interstitial cells. Interestingly, *Soat1* expression was locally enriched within the TLTs, compared to that outside the TLTs (Figure 3C).

### 2.4. SOAT Inhibition Ameliorates TLT Maturation and Renal Fibrosis

To investigate the role of SOAT in TLT formation and maturation in the kidney, SOAT inhibitor avasimibe was mixed with a HFD at 0.01%, and IRI was conducted (Figure 4A). Avasimibe dosage was set at an amount sufficient to lower renal CE levels in mice fed a HFD (Appendix A, Data S2). The body weights of mice fed an avasimibe-containing HFD were comparable to those of mice fed a HFD without avasimibe (Appendix A). On day 35 after IRI, the avasimibe-treated group also exhibited mononuclear cell aggregates around the renal artery, which were predominantly composed of T and B lymphocytes with signs of proliferation (Figure 4B,C). While cumulative TLT size and total TLT number were comparable between the groups, the proportion of advanced stage TLT was significantly decreased in avasimibe-treated mouse kidneys (Figure 4D,E). Avasimibe treatment significantly decreased the mRNA expression levels of *Ccl19*, *Ifng*, and *Tnfa* in injured mice kidneys (Figure 4F). IRI-induced renal fibrosis was also attenuated in avasimibe-treated mice (Figure 4G).

## 3. Discussion

In the present study, we demonstrated that HFD feeding promoted TLT formation and chronic inflammation in injured kidneys. Comprehensive lipidomic analysis revealed CE accumulation in the aged injured kidneys with TLTs. SOAT1, the key enzyme that esterifies cholesterol with fatty acid to produce CE, was significantly increased in aged injured kidneys with TLT, especially within TLTs. Treatment with avasimibe, a SOAT inhibitor, ameliorated TLT maturation and renal fibrosis in mice, suggesting that targeting SOAT1 may represent a novel therapeutic strategy for TLT-related kidney disease.

TLT develops in parallel with the maturation of fibroblasts into the FDCs [7,16], which require TNFα signaling [17]. Indeed, mice lacking TNFα or TNFR lack organized mature CD21-expressing FDCs [17,18]. In our previous study, we demonstrated that, by means of the lineage tracing method utilizing the protein-zero-Cre mouse line, resident fibroblasts in aged kidneys transdifferentiated into FDCs in response to kidney injury [7]. In the present study, *Tnfa* expression was increased in the injured kidneys of HFD-fed mice, and the number of TLTs in advanced stage, with FDC networks, also increased in these kidneys. In addition, avasimibe treatment decreased HFD-induced *Tnfa* expression and the percentage of advanced stage TLTs. Considering that SOAT1 blockade reduces TNFα expression of macrophages and vascular endothelial cells stimulated with cholesterol or LPS [19,20,21], these results suggest that TNFα plays important roles in TLT development promoted by HFD, and that SOAT inhibition ameliorates this process.

LC–MS/MS-based lipidomics enabled the comprehensive profiling of lipids without biases. While there have been several lipidomic studies investigating aged or HFD-administered kidneys [22,23], comprehensive analysis focused on TLT formation has not been conducted. Our lipidomic data demonstrated that, in line with TLT formation, CEs were upregulated in aged injured kidneys. We also revealed that the expression of the CE-producing enzyme SOAT1 was elevated in aged injured kidneys. We showed that SOAT1 is highly expressed in TLTs, suggesting the presence of a TLT-associated niche that facilitates SOAT1 expression inside TLTs. TLTs abundantly produce TNFα and IFNγ [7,24], both of which have been reported to induce SOAT1 expression in macrophages [25,26]. Thus, these cytokines might be involved in the upregulation of SOAT1 expression in aged kidneys. It was also reported that TGFβ1 upregulates SOAT1 expression during differentiation of monocytes into macrophages [27]. SMAD3, downstream of TGFβ1, is predicted to bind to the SOAT1 promoter to upregulate SOAT1 expression [28]. As the TGFβ/SMAD pathway is a major pathway in renal fibrosis in aging, diabetic, and ischemic kidneys [29], this pathway might also be involved in the upregulation of SOAT1 expression in our model.

Previous studies showed the beneficial effects of SOAT1 inhibition in murine disease models, such as atherosclerosis and Alzheimer’s disease [30,31]. In the present study, we also showed that SOAT1 blockade attenuated renal inflammation and fibrosis in IRI models with HFD feeding. Although in situ hybridization showed that *Soat1* signal enrichment was observed within TLTs, *Soat1* signals were also detectable in various renal resident cells. Thus, several mechanisms are assumed to be involved in this improvement. First, TLT-mediated inflammation might be reduced with SOAT1 inhibition. We found the significant reduction in pro-inflammatory cytokine expressions and the number of advanced staged TLTs in avasimibe-treated mouse kidneys. Lymphocytes activated in TLTs produce a large amount of IFNγ, which induces tubular injury by mitochondrial dysfunction and promotes fibrosis [32,33], and SOAT inhibition might ameliorate this process. Second, SOAT1 blockade reduces migration capacity of monocytes/macrophages in inflamed tissue [19], which might contribute to less inflammation and fibrosis. Consistent with this, SOAT inhibition resulted in reduced macrophage/monocyte marker expression in injured kidneys (Appendix A). Third, SOAT1 blockade may improve renal fibrosis and inflammation via the protection of renal resident cells, such as podocytes. Liu et al. showed that CE induced lipotoxicity and contributed to kidney disease progression in 2 murine glomerular injury models, and SOAT1 inhibition reduced the CE in human podocytes [34], although the relative contributions of these mechanisms to kidney disease progression are still unclear. While our results suggest that CE and SOAT are one of the key factors for HFD-induced development of TLT and fibrosis, further studies, including a three-group experiment consisting of control, HFD and HFD added avasimibe-fed IRI mice, is needed to evaluate the contribution of CE and SOAT to the TLT development and fibrosis promoted with HFD.

Several SOAT inhibitors were studied for atherosclerotic diseases or hypercholesterolemia, and are well tolerated in phase II and/or III trials in human [35,36,37]. Our study proposes the further application of SOAT inhibitors in obese and/or aged CKD patients.

In conclusion, we demonstrated that a HFD promoted TLT formation and chronic inflammation in mice kidneys. LC–MS/MS-based lipidomics revealed enhanced SOAT pathway activation in the kidneys with TLTs. Our results provide new insights into obesity and CKD progression associated with TLT development, and provide a rationale for studying the SOAT pathway in CKD. Further studies on SOAT pathways may offer novel therapeutic approaches for CKDs related to metabolic syndrome.

## 4. Materials and Methods

### 4.1. Animals

For the HFD feeding and drug administration, C57BL/6J female mice were purchased from Oriental Yeast (Tokyo, Japan) and maintained at the RIKEN animal faculty under specific pathogen-free conditions. At 28 days prior to surgery, 8-month-old mice were placed on CE-2 (CLEA, Tokyo, Japan) as a normal control chow, HFD-32 (CLEA) as an HFD, or HFD-32 supplemented with 0.01% avasimibe (CI-1101) (A10102; AdooQ, Irvine, CA, USA). A previous report evaluating the effect of HFD-32 compared to CE-2 found no difference in glucose and cholesterol metabolism change or body weight gain between males and females of C57B6/J [15].

For comparison analysis of aged and young mice, 8-week-old and 12-month-old male mice (C57BL/6J) were used as young and aged mice, respectively. Mice were purchased from Japan SLC (Shizuoka, Japan) and maintained under specific pathogen-free conditions in the animal faculty of Kyoto University.

### 4.2. Kidney Ischemic Reperfusion Injury Model

Ischemic reperfusion injury (IRI) was induced as previously described [7]. Briefly, mice were anesthetized and IRI was induced by clamping the unilateral renal pedicles for a certain period of time. 

All animal experiments at Kyoto University were approved by the Animal Research Committee, Graduate School of Medicine, Kyoto University (protocol number: MedKyoto20187), and were conducted in accordance with the Guide for the Care and Use of Laboratory Animals (National Institutes of Health, Bethesda, MD, USA). All animal experiments at RIKEN were performed in full compliance with the RIKEN approved protocols and institutional guidelines. All animal studies were approved by the Animal Care and Use Committee of the RIKEN Yokohama Institute (protocol number: AEY2021-002(5)).

### 4.3. Lipid Extraction

Total lipids from the organs were extracted as previously described [38]. Briefly, kidneys were frozen in liquid nitrogen and homogenized by shaking with metal corn using a Multi-Beads Shocker (YASUI KIKAI, Osaka, Japan) for 15 s at 2500 rpm. Next, 200 μL of methanol per 10 mg of kidney was added and homogenized under identical conditions. Whole lipids were extracted using a single-phase extraction. In brief, 200 μL of the suspension was incubated for 1 h at room temperature, mixed with 100 μL of chloroform, and then incubated again for 1 h at room temperature. Subsequently, 20 μL of water was added, and the samples were incubated for 10 min. After extraction, the samples were centrifuged at 2000× *g* for 10 min, and the supernatants were collected.

### 4.4. Liquid Chromatography-Mass Spectrometry-Based Untargeted Lipidomics

The lipid profile of the kidney was analyzed using LC-MS/MS, as previously described [38]. Briefly, an ACQUITY UPLC system (Waters, Milford, MA, USA) coupled with a quadrupole time-of-flight mass spectrometer (TripleTOF 6600, SCIEX, Framingham, MA, USA) was used. LC separation was performed using a reverse-phase column (ACQUITY UPLC BEH peptide C18; 2.1 × 50 mm, 1.7 µm particle size; Waters) The sample injection volumes were 2 μL and 1 μL for the negative and positive ion modes, respectively.

### 4.5. Annotation and Quantification of Lipids

The untargeted analysis data were analyzed using MS-DIAL software version 4.18 (http://prime.psc.riken.jp/ accessed on 22 March 2020) [39]. Briefly, the raw MS files (WIFF format file) were converted to ABF (analysis base file format) using the freely available Reifycs file converter (http://www.reifycs.com/AbfConverter/ accessed on 1 March 2020) and imported into MS-DIAL. MS/MS spectra of the identified lipids were confirmed and manually corrected as necessary. The quantity was compared based on peak intensity.

Lipidomic data were converted to z-scores, analyzed, and visualized using Multiple Experiment Viewer (MeV 4.9.0) [40].

### 4.6. Quantification of TLT Size

The kidneys of the mice were harvested, cut along the short axis at the maximum area of the whole kidney, fixed in 4% paraformaldehyde/PBS, embedded in paraffin, and sectioned into 3 µm thick slices. Slides were deparaffinized, rinsed, and stained with periodic acid-Schiff. Images were obtained and the TLT size was measured using Keyence BZ-700X.

### 4.7. Evaluation of TLT Stage

The TLT stage was assessed as previously described [12]. TLT identification was based on the unique localization of inflammatory cell aggregates and the presence of B cell aggregates. Quantification of TLT number and stage in the renal cortex was examined by immunofluorescence of (1) CD3 and CD20, and (2) Ki67 and CD21 in two serial sections for each mouse. TLT stage determination was performed based on the presence of CD21-positive FDCs and germinal centers, that is, dense Ki67-positive B cell clusters, within TLTs. TLTs containing neither FDC nor germinal centers were defined as stage I TLTs, whereas TLTs that contained FDCs but lacked germinal centers were defined as stage II TLTs. TLTs with prominent FDCs and germinal centers were classified as stage III. Advanced stage TLTs were defined as TLTs with FDCs (i.e., stage II and III TLTs).

### 4.8. Evaluation of Fibrotic Area

Paraffin-embedded sections (3 µm thick) were deparaffinized, rinsed, and stained for 1 h with Picrosirius red. The slides were then washed twice with 0.5% acetic acid. Histological images were taken of each kidney section at the cortical and cortico-medullary fields, except for the TLT area (*n* = 3). All images were obtained by a Keyence BZ-700X at 200× magnification using the same laser power and gain intensity. The fibrotic area was quantified using image analysis software (FIJI, ImageJ, NIH, Bethesda, MD, USA) [41]. The pictures were split into red, blue, and green. The green color picture was subtracted from the red color to obtain the Picrosirius red-positive region.

### 4.9. Immunofluorescence

Immunofluorescence analysis was performed as previously reported [7]. The obtained kidneys were fixed in 4% paraformaldehyde/PBS, dehydrated with PBS containing sucrose, and embedded in O.C.T. compound.

Frozen blocks were cryosectioned into 6 µm thick slices. Sections were blocked with ImmunoBlock (CTKN001; KAC, Kyoto, Japan) or 3% BSA for 1 h and incubated overnight at 4 °C with primary antibodies. Sections were then stained with the appropriate secondary antibodies.

The following antibodies were used: anti-Ki67 (16667; Abcam, Cambridge, UK, 1/200), -CD21 (ab75985; Abcam, 1/400), -p75NTR (AF1157; R&D Systems, Minneapolis, MN, USA, 1/100), -B220 (103254; BioLegend, San Diego, CA, USA, 1/100), and -CD3 (100212; BioLegend, 1/50).

### 4.10. Quantitative Reverse Transcription Polymerase Chain Reaction (RT-qPCR)

RNA was isolated using the RNeasy kit (QIAGEN, Hilden, Germany), and cDNA synthesis was performed using Prime Script RT Master Mix (Takara, Shiga, Japan). RT-qPCR was performed by 40-cycle amplification using a gene-specific primer and TB Green Premix Ex Taq kit (Takara) on the Step One Plus Real-Time PCR system (Applied Biosystems, Waltham, MA, USA). Relative gene expression values were normalized to endogenous *Gapdh* levels. Relative fold changes in expression were calculated using the ∆∆CT method. The copy number of sterol O-acyltransferase 1 or 2 (*Soat1* or *Soat2*) in the samples was calculated using a standard curve that was constructed by 10-fold dilutions of the specific plasmid containing each sequence. The primer sequences used are listed in Appendix A.

### 4.11. In Situ Hybridization

Detection of mouse *Soat1* RNA was performed on the abovementioned cryosections and formalin fixed paraffin embedded sections of the mice kidneys subjected to 45 min IRI on day 30 using Advanced Cell Diagnostics (Hayward, CA, USA) RNAscope (R) Multiplex Fluorescent Reagent Kit v2 (#323110) and RNAscope Target Probes Mm-*Soat1* (#541821) according to the manufacturer’s instructions.

### 4.12. Statistics and Data Visualization

Data are reported as the mean ± S.E.M. Statistical significance was assessed using two-tailed student’s *t*-test for comparisons between the two groups and using Tukey’s HSD test for comparison of multiple groups using the R programming language. Graphs were also visualized using the R programming language.

Lipidomic data were analyzed with one-way analysis of variance (ANOVA) test using Multiple Experiment Viewer (MeV 4.9.0) [40]. Statistical significance was set at *p* < 0.05.

## Figures and Tables

**Figure 1 ijms-23-15465-f001:**
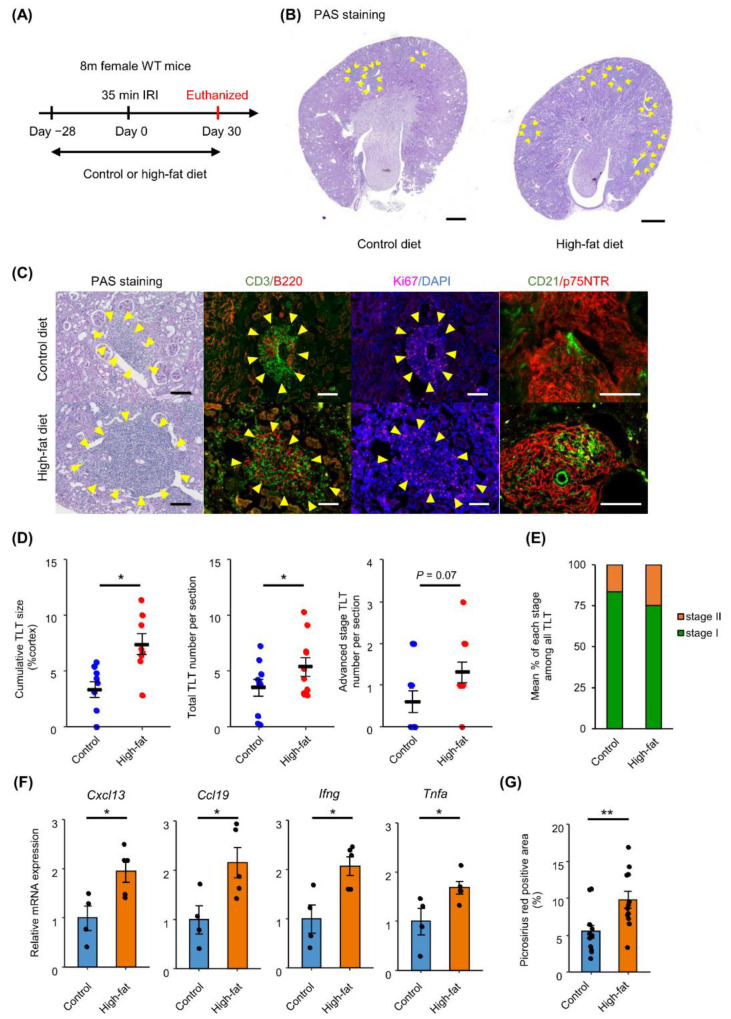
High-fat diet feeding promotes TLT formation in kidneys. (**A**) Experimental protocol. Wild-type (WT) 8-month-old female mice fed a control or high-fat diet (HFD) from 28 days prior to ischemic reperfusion injury (IRI). Diet was fed for 30 days after IRI. (**B**) Periodic acid–Schiff (PAS) staining of kidneys 30 days after 35 min IRI under control diet and under HFD. Arrows indicate TLTs. (**C**) PAS staining and immunofluorescence of TLTs under control diet and HFD. B220, CD3, Ki67, and DAPI were stained on the same section. CD21 and p75NTR were stained inside TLTs. Yellow arrows indicate TLT. (**D**) Cumulative TLT size per cortex (control diet *n* = 10, HFD *n* = 8), total TLT number per section (*n* = 10), and advanced stage (stage II or III) TLT number per section (*n* = 10). (**E**) Percentage of advanced stage TLTs detected in each sample (*n* = 10). (**F**) Relative mRNA expressions of *Cxcl13*, *Ccl19*, *Ifng*, and *Tnfa*. Expressions are normalized with *Gapdh* expression and values are presented relative to control kidneys (control *n* = 4, HFD *n* = 5). (**G**) Quantification of Picrosirius red-positive fibrotic area comparing control-fed and HFD-fed mice after IRI (*n* = 12). Values are mean ± S.E.M. Data were analyzed using student’s *t*-test, * *p* < 0.05, ** *p* < 0.01. Scale bars: (**B**) 500 µm, (**C**) 100 µm. TLT stages were defined as follows. Stage I: TLT lacking either CD21+ follicular dendritic cells (FDC) or germinal center, stage II: TLT containing FDC but lacking germinal center, stage III: TLT containing both FDC and germinal center, based on the criteria described in [12].

**Figure 2 ijms-23-15465-f002:**
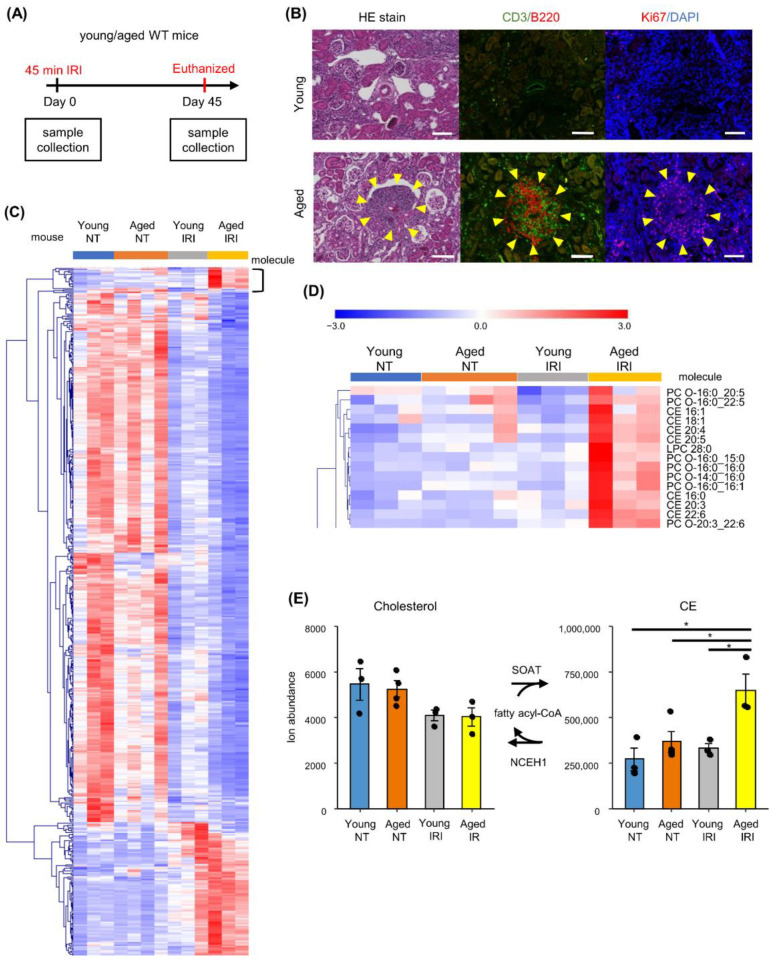
Untargeted lipidomics of aged and young mice kidneys with/without IRI. (**A**) Experimental protocol of lipidomic analysis. (**B**) Hematoxylin and eosin staining and immunofluorescence of kidney section from young (8-week-old) and aged (12-month-old) mice after 45 min IRI. Yellow arrows indicate TLT. Scale bars: 100 μm. (**C**) Cluster analysis of molecules significantly changed among groups (*n* = 3–4 for each group) by one-way analysis of variance (*p* < 0.05, 511 molecules). Intensity of each molecule was converted to z-score and displayed as a heatmap. Detailed data of clustered molecules are reported in Appendix A. (**D**) Cluster that contains molecules uniquely increased in aged kidneys after IRI. (**E**) Cholesterol and cholesteryl ester (CE) abundance measured by LC–MS/MS. Statistical significance was analyzed with Tukey’s HSD test, * *p* < 0.05. Abbreviations: NT non-treated, CE cholesteryl ester, PC-O ether-linked phosphatidylcholine. LPC lysophosphatidylcholine, SOAT sterol O-acyltransferase, NCEH neutral cholesterol ester hydrolase.

**Figure 3 ijms-23-15465-f003:**
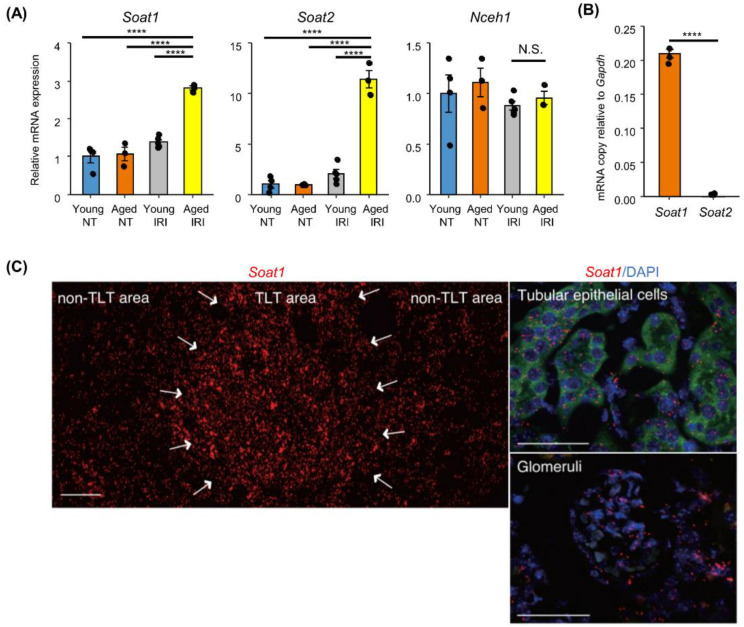
*Soat* expression in kidneys with TLT. (**A**) Relative mRNA expression of CE metabolizing enzymes. Young (8-week-old) and aged (12-month-old) mice 45 days after 45 min IRI or without IRI (*n* = 3–5) were compared. Data are normalized with *Gapdh* and presented as fold change relative to young non-treated (NT) kidneys. (**B**) Comparison of *Soat1* and *Soat2* mRNA expressions in aged kidneys after IRI. The mRNA copy numbers per sample were obtained and normalized with *Gapdh* copy number per sample. (**C**) In situ hybridization of *Soat1* (red) in kidneys 45 days after 45 min IRI. Arrows indicate TLT. In the image of tubular epithelial cells, the green staining represents background autofluorescence that was adjusted to show tubular structure. Scale bars: 50 μm. Values are mean ± S.E.M. Statistical significance was analyzed with (**B**) Tukey’s HSD test, and (**C**) student’s *t*-test. **** *p* < 0.0001. N.S. indicates not significant.

**Figure 4 ijms-23-15465-f004:**
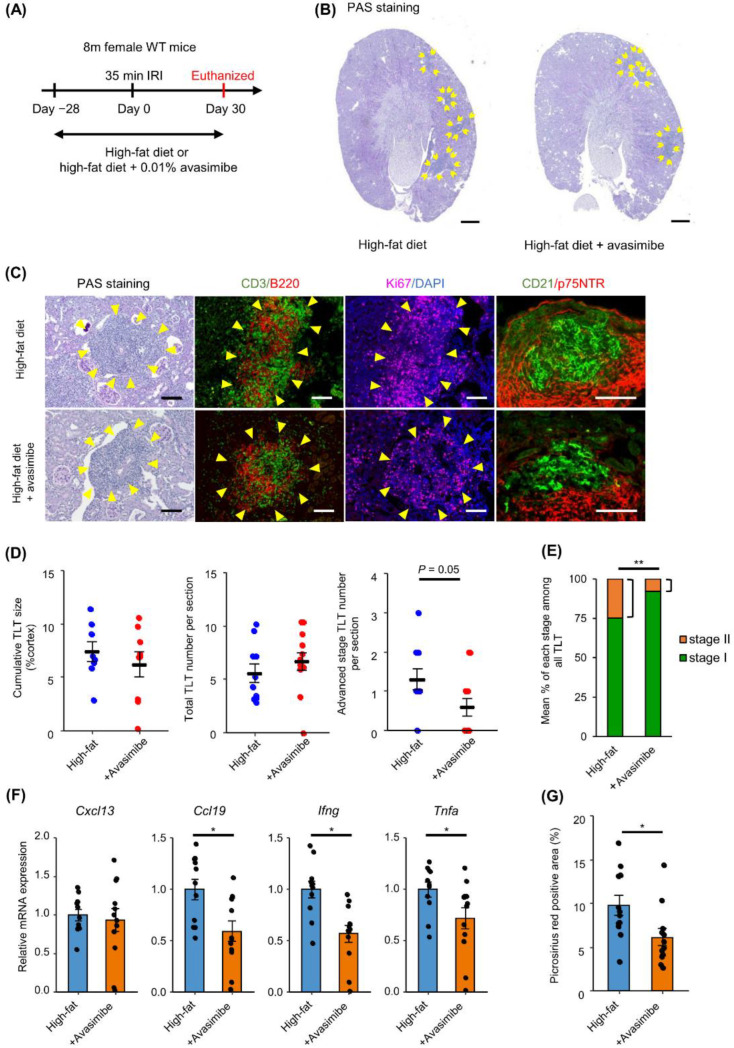
SOAT inhibitor does not significantly decrease TLT size, but inhibits TLT maturation, decreases inflammatory cytokines, and ameliorates fibrosis. (**A**) Experimental protocol. Eight-month-old female mice administered a HFD or HFD + 0.01% avasimibe diet for 28 days and induced 35 min IRI. Kidneys were harvested 30 days after IRI. (**B**) PAS staining of kidneys 30 days after 35 min IRI under HFD, and HFD + avasimibe. Arrows indicate TLTs. (**C**) PAS staining and immunofluorescence of TLTs under HFD and HFD + avasimibe. B220, CD3, Ki67, and DAPI were stained on the same section or serial section. CD21 and p75NTR were stained inside TLTs. Yellow arrows indicate TLT. (**D**) Cumulative TLT size per cortex (HFD *n* = 8, HFD + avasimibe *n* = 9), total TLT number per section (HFD *n* = 10, HFD + avasimibe *n* = 12), and advanced stage TLT number per section (HFD *n* = 10, HFD + avasimibe *n* = 12). (**E**) Percentage of advanced stage TLTs detected in each sample. HFD *n* = 10, HFD + avasimibe *n* = 12. (**F**) Relative mRNA expressions of *Cxcl13*, *Ccl19*, *Ifng*, and *Tnfa*. Expressions are normalized with *Gapdh* expression and values are presented relative to HFD-fed kidneys. (HFD *n* = 11, HFD + avasimibe *n* = 12). (**G**) Quantification of Picrosirius red-positive fibrotic area comparing HFD- and HFD + avasimibe-fed mice after IRI (*n* = 12). Values are mean ± S.E.M. Data were analyzed using student’s *t*-test, * *p* < 0.05, ** *p* < 0.01. Scale bars: (**B**) 500 µm, (**C**) 100 µm.

## Data Availability

The datasets analyzed for this study can be found in the MetaboLights (www.ebi.ac.uk/metabolights/index) under accession number MTBLS4988 and MTBLS4994. The datasets are also found in the DROPMet server under accession number DM0046.

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
