# Peer review of "Sterol O-Acyltransferase Inhibition Ameliorates High-Fat Diet-Induced Renal Fibrosis and Tertiary Lymphoid Tissue Maturation after Ischemic Reperfusion Injury"

_ijms, 2022, doi:10.3390/ijms232415465_

Round 1

Reviewer 1 Report

Remarks to the author:

In this study Ariyasu et al have investigated the relationship between high fat diet (HFD) induced metabolic disorders and renal tertiary lymphoid tissues (TLT) formation and subsequent CKD development after AKI. Control diet and HFD fed female mice at 8 months of age has been used in the study to compare the dietary effects on ischemic reperfusion injury whereas 8weeks and 12 months old male mice were used for age comparison to evaluate TLT lesions after an ischemic reperfusion injury. The authors have reported that HFD promote renal TLT formation and inflammation via sterol O-acyltransferase SOAT) 1-dependent mechanism and treatment with avasimibe, a SOAT inhibitor, attenuates TLT maturation and renal inflammation in HFD-fed mice subjected to IRI. Thus, the authors conclude that SOAT1-dependent CE accumulation plays an important role in TLT formation in CKDs.

Specific comments:

1. Materials and methods:

i) For comparison analysis of aged and young mice, the authors have used male mice and, for the HFD feeding and drug administration, they have used the female mice. There are hormonal differences (sex hormones) in two genders and even the females have changes in their sex hormones within their oestrus cycle which affects the body metabolism and susceptibility of different disease conditions. Therefore, a clear justification is needed to address the hormonal issues those can affect when comparing the outcomes using different genders.

2.         Results:

i) The authors have conducted the experiments using female mice. Were there any experiments carried out to see whether female animals show any differences compared to male mice in relation to their hormonal changes in which might affect the metabolism?If so, those results need to be discussed.

ii)  Figure 1E – The figure shows stage I and stage II data, whereas in the text (line 89), the authors have mentioned that they have looked at the advanced stages II and III lesions in the animals. Also, it would be informative if the staging system (criteria) that has been used for the study is mentioned or referred.

iii) When analysing the data, for kidney cholesterol and Kidney CE, the authors have compared between different diet groups (Figure S2) in addition to HFD and HFD + 0.01% avasimibe groups. Similarly, when interpreting the data on effects of SOAT inhibition on TLT maturation and renal fibrosis, the data on control diet group would be necessary to confirm whether LTL size and number, and renal fibrosis has attenuated or reversed compared to control diet fed IRI mice (Figure 4).

3.         Discussion and Conclusion:

i)  Authors need to be consistent with abbreviations (eg:TNFa) throughout.

Reviewer 2 Report

Ariyasu et al. investigated the effects of sterol O-acyltransferase inhibition in high-fat diet (HFD)-induced renal inflammation, fibrosis, and tertiary lymphoid tissue (TLT) maturation after ischemia/reperfusion (I/R) injury in female mice. Their main findings are that i) HFD feeding promoted TLT formation and chronic inflammation after I/R injury in kidneys; ii) CE accumulated and the key enzyme SOAT1 upregulated in aged I/R injured kidneys in TLTs; iii) treatment with avasimibe, a SOAT inhibitor, ameliorated TLT maturation and renal fibrosis.

The study seems to be carefully designed and the authors used plenty of techniques. I have only minor comments:

1. Why only female mice are used? Please discuss it?

2. Do the authors have previous results on TLT formation induced by HFD without I/R injury? Please discuss it.

3. Based on literature data, SMAD3 binding region in the SOAT1 promoter is predicted by Genomatix. Are there any data on the relationship between TGF-beta/SMAD pathway and SOAT1 in kidney? 
